# Physical Activity during Pregnancy and Newborn Body Composition: A Systematic Review

**DOI:** 10.3390/ijerph19127127

**Published:** 2022-06-10

**Authors:** Brenna R. Menke, Cathryn Duchette, Rachel A. Tinius, Alexandria Q. Wilson, Elizabeth A. Altizer, Jill M. Maples

**Affiliations:** 1School of Occupational Therapy, University of Indianapolis, Indianapolis, IN 46227, USA; menkeb@uindy.edu; 2Alabama College of Osteopathic Medicine, Dothan, AL 36303, USA; cathryn.duchette@gmail.com; 3School of Kinesiology, Recreation and Sport, Western Kentucky University, Bowling Green, KY 42101, USA; rachel.tinius@wku.edu; 4Preston Medical Library, University of Tennessee Graduate School of Medicine, Knoxville, TN 37920, USA; aqwilson@utmck.edu; 5Department of Obstetrics and Gynecology, University of Tennessee Graduate School of Medicine, Knoxville, TN 37920, USA; eaaltizer@utmck.edu

**Keywords:** exercise, infant anthropometrics, neonatal adiposity, maternal physical activity

## Abstract

The current literature demonstrates that not only is exercise during pregnancy safe, but it has substantial maternal and infant benefits and appears to influence infant growth/size throughout pregnancy and at birth. However, many existing studies have investigated only the effects of prenatal exercise on birth weight. The purpose of this review was to determine the impact or association of maternal physical activity during pregnancy on neonatal body composition assessed between birth and two weeks of age. Electronic database searches were conducted on 29 July 2019 for randomized control trials and cohort studies, with an updated search completed on 8 January 2021. A total of 32 articles that met eligibility criteria were selected for review. Overall, prenatal exercise was not associated with infant body composition at birth. Yet, five of the studies identified suggest that infant body composition could be influenced by higher volumes of mid-to-late term prenatal physical activity. This systematic review was conducted according to the Preferred Reporting Items for Systematic Reviews and Meta-Analyses (PRISMA) guidelines and registered in PROSPERO (Registration No. CRD42020160138).

## 1. Introduction

Scientific evidence has demonstrated the safety and efficacy of exercise during the perinatal period [1,2,3,4,5]. Physical activity (PA) during pregnancy provides many maternal health benefits, including improved glucose control, lower gestational weight gain, lower systemic inflammation, reduced risk for preeclampsia, reduced risk for operative deliveries, and faster postpartum recovery time [1,2,3,4,5]. In addition to improvements in maternal health, physical activity during pregnancy has also been shown to benefit the neonate, including healthy growth and improved cognition and intelligence [6,7]. Specifically, exercise during pregnancy has been linked to lower and healthier infant birth weight (without increasing risk for low birth weight) [8,9].

Birth weight has been used historically to indicate a healthy intrauterine growth environment; however, birthweight is not a strong predictor of important future health outcomes [10], particularly at the individual level [11]. Despite the lack of evidence to support its connection to downstream outcomes, it is widely utilized clinically and is typically one of the first assessments of a newborn. One reason why birth weight may be a poor predictor of outcomes is it cannot adequately estimate fat mass relative to fat-free mass in infants. It also does not account for how much an infant “should have weighed” based on their length and genetic potential [11]. Assessing body composition at birth (i.e., fat mass versus fat-free mass) may provide additional details indicating healthy fetal/neonatal growth/size. Further, adiposity levels at birth are a better predictor of metabolic syndrome and other non-communicable diseases later in life than birthweight alone [12].

Taken together, understanding the relationship between physical activity during pregnancy and infant body composition is essential. Exercise during pregnancy appears to influence infant growth/size [9,13]; however, many existing studies have investigated only the effects of prenatal exercise on birth weight and not body composition. Over the past 10 years, research investigating the role of exercise during pregnancy on infant body composition has gained momentum. However, the existing evidence on the impact physical activity during pregnancy has on infant body composition is unclear and oftentimes conflicting. The conflicting results may be due to how physical activity was assessed, what timepoint during pregnancy exercise was studied, and the frequency, intensity, time, and type of exercise performed. The purpose of this systematic review is to determine the association of maternal physical activity during pregnancy and neonatal body composition assessed between birth and two weeks of age.

## 2. Methods

This systematic review was conducted according to the Preferred Reporting Items for Systematic Reviews and Meta-Analyses (PRISMA) guidelines. The review protocol was pre-registered with the International Prospective Registrar of Systematic Reviews (PROSPERO) registration number CRD42020160138. We followed the PRISMA reporting guidelines (see Appendix B).

### 2.1. Eligibility Criteria

#### 2.1.1. Types of Study Designs

Eligible study designs included randomized controlled clinical trials (RCTs), prospective cohort studies, and retrospective cohort studies that assessed physical activity levels at any point during pregnancy and assessed infant body composition within two weeks of delivery. Studies were included regardless of intensity, duration, or mode of physical activity utilized. Only studies that were peer-reviewed with scientific credibility were included. The decision to include non-RCTs was based on several factors, including lack of existing RCTs assessing infant body composition as an outcome; poor adherence to interventions, making it difficult to truly evaluate the relationship between physical activity levels and infant outcomes; and the idea that other study designs may better allow assessment of physical activity as a continuous variable versus as a grouping variable (i.e., control group vs. exercise group).

#### 2.1.2. Types of Participants

Women of all pre-pregnancy body mass indices (BMIs) were included. Studies including women with gestational diabetes and preeclampsia were excluded. These specific conditions have known and particular effects on infant body size/composition and would make it difficult to discern the impact of physical activity on infant outcomes. Studies including multiple gestation pregnancies were also excluded.

#### 2.1.3. Types of Outcome Measurements

For the mode of exercise, studies including all modes of physical activity were included (aerobic training, resistance training, combination training). The physical activity assessment could be through self-reporting (e.g., the Pregnancy Physical Activity Questionnaire), compliance to an intervention, or objective assessments (i.e., accelerometer, pedometer, or doubly-labeled water). Exercise data could be collected during any timepoint of pregnancy (1st, 2nd, 3rd trimester, or throughout pregnancy). For the outcome of infant body composition, the study had to include an assessment or estimate of infant adiposity (and not just birth weight). These included ponderal index, BMI, abdominal circumference, or body fat percentage from bioelectrical impedance, dual X-ray absorptiometry (DXA), skinfold anthropometry, or air-displacement plethysmography. Any studies including an initial assessment of body composition on infants >2 weeks of age were excluded.

### 2.2. Search Strategy

Electronic searching of PubMed/MEDLINE, EMBASE (Elsevier), Web of Science Core Collection (Thomson Reuters), and CINAHL (EBSCO) databases took place on 29 July 2019. Phrases and controlled vocabulary headings for each component of the population and outcome framework (PICO) were combined using OR and then using AND (exercise/physical activity, pregnancy, neonate, and body composition). The search strategy was first created in PubMed (See Appendix C) and then translated for each database platform as appropriate. Manual searches of reference lists were conducted on all eligible articles following the screening. An updated search was completed on 8 January 2021.

### 2.3. Assessment of Bias

The risk of bias (RoB) was assessed on all studies selected for inclusion. Randomized control trials (RCTs) were evaluated using Version 2 of the Cochrane risk-of-bias tool (RoB 2) [14]. Each RCT was assessed using the study’s “per-protocol” effect by independent reviewers (CD, BM, EA). Discrepancies in RoB scoring were resolved by discussion between reviewers and an additional study team member (JM). Five bias domains were assessed (i.e., selection, performance, detection, attrition, and reporting), and an overall bias score was calculated.

RoB in case-control and single-arm cohort studies was evaluated using a modified Newcastle-Ottawa Scale (NOS) [15] (See Appendix D). Again, discrepancies in RoB scoring were resolved by discussion between reviewers and an additional study team member (JM). Three bias domains were assessed (i.e., selection, comparability, exposure/outcome), and an overall bias score was calculated (See Appendix E).

### 2.4. Data Management

All articles retrieved from the electronic databases were imported to EndNote, and duplicates were removed. After deduplication, titles and abstracts were uploaded to Rayyan, a web and mobile app for systematic reviews, for screening [16]. In stage one of screening, reviewers independently screened titles and abstracts of articles yielded by the search to identify potentially eligible studies based on the inclusion criteria (CD, BM). In stage two, if eligibility was unclear from the review of title and abstract, the full text was obtained for further assessment, and discrepancies were resolved by discussion between reviewers and consensus (CD, BM, JM, RT). In stage three of the screening process, full versions of relevant articles were obtained and carefully assessed to ensure they fit the predetermined study criteria. The full-text screening was conducted by two independent reviewers (CD, BM). Discrepancies were resolved by consensus among the study team (CD, BM, JM, RT).

### 2.5. Data Extraction

Extracted data included year published, study design, sample size, ethnicity, pre-pregnancy weight status, time point of pregnancy, physical activity assessment (frequency, intensity, time, type), infant body composition assessment, timepoint for infant body composition assessment, and main results found. Data from each study were extracted by one reviewer (BM) and checked by study team members (CD, RT, JM). All extracted data were organized by design type and risk of bias score, as shown in Table 1, Table 2, Table 3, Table 4, Table 5, Table 6, Table 7, Table 8 and Table 9.

## 3. Results

The PRISMA flow chart (Figure 1) shows the number of articles at each stage of the screening process. Database searches identified a total of 591 studies, and an additional 17 studies were identified through other sources. An updated search was completed on 8 January 2021, and a further 79 studies were identified. Duplicates were removed, and of the 554 studies screened, 471 were excluded based on title, abstract, or outcomes measured. The remaining 83 studies were further evaluated based on full text, and 51 articles were excluded for wrong publication type (*n* = 26) (i.e., systematic review, study protocol, cohort characteristics report, conference presentation, book), wrong study design (*n* = 7), and wrong outcome (*n* = 23) (i.e., ineligible infant measurements); 32 articles were included for evaluation. Of these, 13 were randomized controlled trials [17,18,19,20,21,22,23,24,25,26,27,28,29], 16 were single-arm cohort studies [30,31,32,33,34,35,36,37,38,39,40,41,42,43,44,45], and 3 were case-control studies [5,6,46].

Overall, prenatal exercise was not associated with infant body composition at birth. Of the 32 articles included, only five suggested a difference in infant body composition could be influenced by prenatal physical activity [20,33,34,36,46]. Specifically, those five studies found that women with a greater total volume of mid-to-late term physical activity gave birth to infants with less fat mass than those who had little to no PA throughout pregnancy.

### Risk of Bias Assessment

The overall risk of bias for randomized controlled trials was low to- moderate. Approximately 54% of the studies (7 of 13) were rated as having “some concerns” for the overall potential risk of bias [21,22,24,25,27,28,29] (Figure 2). This rating was primarily due to the seven studies being rated as having “some concerns” in the performance risk of bias domain, which was assessed by rating deviations from the intended intervention. There was a low risk of selection bias for all studies, as all studies reported a random allocation sequence. Only one study did not provide enough information regarding the allocation sequence being concealed until after participants were enrolled/assigned to the intervention [23]. A summary of the RCT RoB scoring is listed in Figure 2.

For case-control and cohort studies, the risk of bias was generally low to moderate. We did not detect strong evidence of publication bias in these studies (See Appendix E).

## 4. Discussion

Overall, this systematic review suggests maternal PA during pregnancy has minimal implications on neonatal adiposity, except for extremely high volumes of PA. However, due to the variability in numerous elements of the study design (e.g., PA assessment method, neonatal adiposity assessment method, and timing of data collections), it remained a challenge to synthesize findings. These data suggest that maternal physical activity does not appear to have a negative or detrimental impact on the body composition of the neonate, which is an important conclusion given fear of harm to the unborn child is a factor that prevents many women from getting or staying active during pregnancy [48]. In this review, we report that five studies noted a relationship between maternal PA and infant body composition [20,33,34,36,46]; however, none of them reported utilizing rigorous methods for assessing both maternal PA and neonatal body composition.

### 4.1. Summary of Studies That Showed Increased Maternal PA Is Associated with Decreased Neonatal Adiposity

Only one of thirteen randomized clinical trials had findings indicating that maternal PA has an impact on neonatal anthropometrics [20]. Specifically, findings from this study indicate that previously active women that decrease their physical activity levels by 67% (below the recommended 150 min/week) starting at mid-pregnancy through delivery, had heavier babies at birth (3.9 kg) with a higher amount of body fat (12.1%) compared to women who maintained relatively moderate to high levels of physical activity (at least 150 min/week) from mid-pregnancy to delivery (3.4 kg, 7.9%). Given that RCTs are the highest level of evidence, and 12 of 13 did not detect significant differences in neonatal adiposity between active and inactive women (with a variety of activities and intervention protocols), these collective findings suggest physical activity has a minimal impact on offspring adiposity.

Out of the 16 single-arm cohort studies that were identified, three studies found that maternal physical activity levels were associated with smaller amounts of neonatal adiposity [33,34,36]. In the most recent study from Collings et al., 2020, the authors found that among British white women there was a negative dose-response relationship. Women with moderate to high levels of maternal physical activity during mid-pregnancy (150 min or more per week of moderate intensity exercise) were associated with smaller infant skinfold measurements. However, this relationship was not apparent in the portion of the cohort that identified as ethnic minorities.

The second cohort study by Dahly et al., 2018, determined that the prevalence of babies born with very high fat mass was lowest among women that engaged in frequent moderate-intensity exercise early in pregnancy [34]. While neonatal body composition was assessed with air displacement plethysmography, maternal physical activity was subjectively self-reported with a one-question survey at 15 weeks gestation. This is problematic, as oftentimes physical activity levels change throughout pregnancy [49]. In addition, the maternal PA data were collected with a non-validated single question with only three possible responses (none, some (1–3 times per week) or often (4+ times per week)), which may lead to a lack of sensitivity for detecting important relationships.

The third cohort study, by Harrod et al., 2014, found that increasing levels of maternal physical activity, particularly during late pregnancy, are negatively correlated to neonatal adiposity [36]. In this cohort, women with high levels of late-pregnancy activity had an increased likelihood of SGA compared to women with low levels of physical activity. The authors suggest that the effect could be attributed to a decrease in neonatal fat mass, because there were no differences in neonatal fat-free mass stratified by total energy expenditure quartiles.

Out of three case-control cohort studies included, one study found a relationship between maternal physical activity and neonatal body composition [46]. Clapp and Capeless, 1990, compared infant outcomes between conditioned recreational runners and aerobic dancers (who maintained activity at or above 50% of preconception levels) and matched inactive controls. Infants born to active women had lower birth weight, ponderal index, two-site skinfold thickness, and calculated percent body fat, leading to the conclusion that continuation of regular aerobic exercise results in a reduction in neonatal fat mass. Specifically, the authors reported that differences in neonatal fat mass explained over 70% of the difference in birth weight [46].

### 4.2. Accuracy of PA Assessment (i.e., Type of Assessment, Timing, and Intensity)

Although a cohort design allows for free-living PA to be reported and/or recorded in addition to sports and exercise leisure-time PA, a randomized control trial that implements an intervention may be able to pinpoint specific PA modifying factors (i.e., duration, frequency, type, and intensity) and determine a direct causal relationship between exercise and infant body composition. It is plausible that specific modes, intensities, and/or volumes can have differential effects on infant body composition. Once again, this poses significant challenges to drawing broad conclusions about the relationship between general physical activity and infant adiposity.

While the Pregnancy Physical Activity Questionnaire (PPAQ) is commonly used across the literature, it has been found to overestimate leisure-time PA across all activity categories [50]. The PPAQ also only provides subjective/self-reported exercise information that is subject to recall bias. The accelerometer is the gold standard for objectively measuring free-living PA [51] and is the most reliable and accurate method for assessing MVPA during pregnancy [52]. Because only eight studies utilized objective accelerometry, conclusions regarding the role of PA on infant outcomes must be drawn carefully. In addition to the measurement of PA, the studies included in the review contained a variety of exercise modes (e.g., running, walking, dancing, swimming, cycling, weightlifting), once again posing a challenge in drawing definitive conclusions across studies, as different modes and intensities may have different implications for fetal growth.

Another factor that warrants consideration is the timepoint in pregnancy at which maternal physical activity assessments are taken. Some studies focus on early pregnancy, others on late pregnancy, and some assess throughout (See Table 7, Table 8 and Table 9). Upon careful examination, it is clear that the gestation period in which exercise occurs can impact the interpretation of the relationship between maternal activity levels and neonatal adiposity. Several studies reported late pregnancy activity levels [36,46], while others measured activity early on [34]. Once again, these differences in study design make it challenging to synthesize results, as maternal–fetal physiology changes drastically throughout pregnancy. Therefore, it is feasible that physical activity can impact fetal growth differently depending on the timepoint of pregnancy. For example, pregnant women become more insulin resistant as pregnancy progresses [53]; thus, fuel and substrate utilization change dramatically throughout a pregnancy [54]. The timing of pregnancy in which exercise occurs most likely confounds the relationship between exercise and infant adiposity.

### 4.3. Accuracy of Infant Anthropometric Assessments (i.e., Type of Assessment and Timing)

Another aspect of study design that may play a role in interpreting the relationship between maternal PA and infant anthropometrics is the accuracy of the infant assessment. The majority of included studies utilized skinfolds and/or ponderal indices, both of which are considered poor and imprecise measures of assessing infant body composition [55]. They are often used due to low cost and feasibility; however, it is clear that more accurate assessments of infant body composition (e.g., air displacement plethysmography) are needed in order to design high-quality studies. In addition to the type of anthropometric assessment tool being used, studies often vary in the timepoint at which they assess infant anthropometrics. Varying the timepoint of assessment can be problematic when comparing across studies, as infant size changes rapidly during early development [56]. For example, once the infant starts receiving either breastmilk or formula, that difference in dietary composition alone can impact infant growth patterns [57]. In order to minimize the impact of the infant’s external environment and nutrient consumption (which can make it nearly impossible to draw any sort of conclusion about the direct impact of physical activity during pregnancy on infant adiposity), studies should attempt to take newborn assessments within the first 48 h of life. However, when observing the findings in Table 7, Table 8 and Table 9, many studies assessed infant adiposity around 2 weeks of age.

### 4.4. Is There an Optimal Amount of Physical Activity to Ensure Healthy Neonatal Adiposity?

Evidence suggests that only extreme levels of physical inactivity/activity seem to influence neonatal adiposity. Specifically, higher levels of maternal physical activity and vigorous maternal physical activity, particularly when performed throughout late pregnancy, are consistently associated with decreased neonatal fat mass. There may be a “threshold” point where moderate exercise is beneficial but heavy exercise is detrimental, at least with respect to birth weight [58]. It is unclear what the threshold is at this point. This does have some clinical utility in terms of being able to appropriately counsel the increasing number of women that wish to maintain high levels of physical activity during pregnancy. An increasing number of elite female athletes are choosing to enter motherhood during their competitive years. In fact, a recent consensus statement suggests that since the fertile age of many athletes overlaps with timing for peak performance, adequate information about the implications of exercise during pregnancy is needed so women can make informed decisions [59]. This systematic review adds to the body of literature that suggests pregnant women can feel reassured that they can continue exercise, but depending on the activity, may have to make small adjustments in intensity and volume in order to maintain appropriate fetal growth [59].

It is also possible that a minimal level of physical activity exists to prevent overgrowth. In fact, moderate maternal physical activity promotes appropriate fetal growth. For example, work from Barakat et al. suggests that light intensity resistance exercise training performed during the second and third trimester might attenuate the adverse consequences of maternal body weight before pregnancy on neonatal anthropometrics [17]. Specifically, Barakat et al. found that maternal body weight was positively and significantly associated with birthweight in the control group, but there was no association in the exercise training group. Others have reported that maternal pre-pregnancy BMI and gestational weight gain are positively and independently associated with neonatal adiposity [60]. It is possible that the impact of maternal physical activity on neonatal adiposity may differ according to maternal pre-pregnancy weight status and/or gestational weight gain status.

## 5. Conclusions

Despite the inherent limitations discussed above, which limit the ability to draw conclusions across studies, some important findings were noted. Collectively, the studies suggest that continuing a regular physical activity regimen during pregnancy does not compromise fetoplacental growth. This is critically important, as many women do not engage in appropriate levels of PA due to fear of harm to the unborn child [48]. Pregnant women should be counselled to continue or begin physical activity regimens, as it appears that physical activity (at reasonable intensities and volumes) has minimal effects on infant body composition.

Given the ongoing obesity epidemic, there is a need to identify children at risk of developing obesity as early in life as possible [48]. It is a challenge to identify neonates at risk as there are no published norms for “appropriate” or “ideal” levels of neonatal body fat percentage and/or fat mass (as exist for adults). Because of this, it is difficult to determine “ideal” amounts of PA as it relates to the impact of PA on infant body composition. However, given there were no clear trends of maternal PA on infant body fat percentage among the studies included in this review, it seems reasonable to conclude that PA during pregnancy, at most any level, is beneficial to the mother without unfavorably compromising fetal growth. With PA during pregnancy having substantial maternal and neonatal benefits [1], results from this review add to the consensus that health care providers should continue to encourage PA among patients while dispelling myths that PA could be harmful to the growth of the baby.

## Figures and Tables

**Figure 1 ijerph-19-07127-f001:**
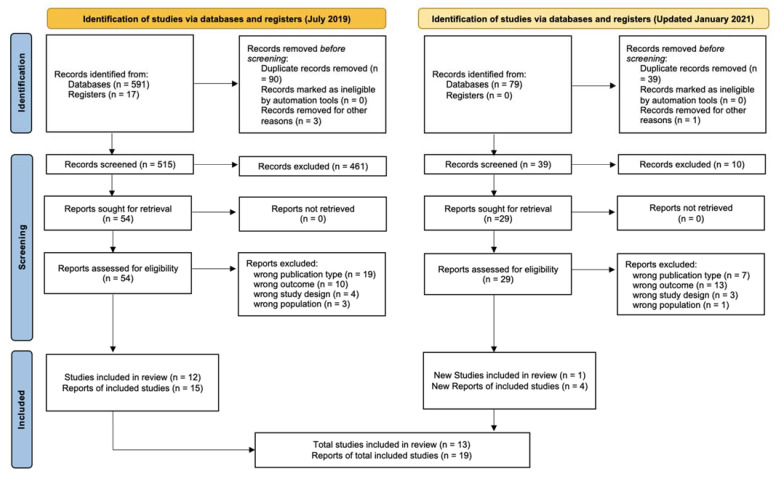
PRISMA flow diagram. Modified from Page, M.J.; McKenzie, J.E.; Bossuyt, P.M.; Boutron, I.; Hoffmann, T.C.; Mulrow, C.D.; Shamseer, L.; Tetzlaff, J.M.; Akl, E.A.; Brennan, S.E.; et al. The PRISMA 2020 statement: An updated guideline for reporting systematic reviews. *BMJ*
**2021**, *372*, n71. doi:10.1136/bmj.n71 [47].

**Figure 2 ijerph-19-07127-f002:**
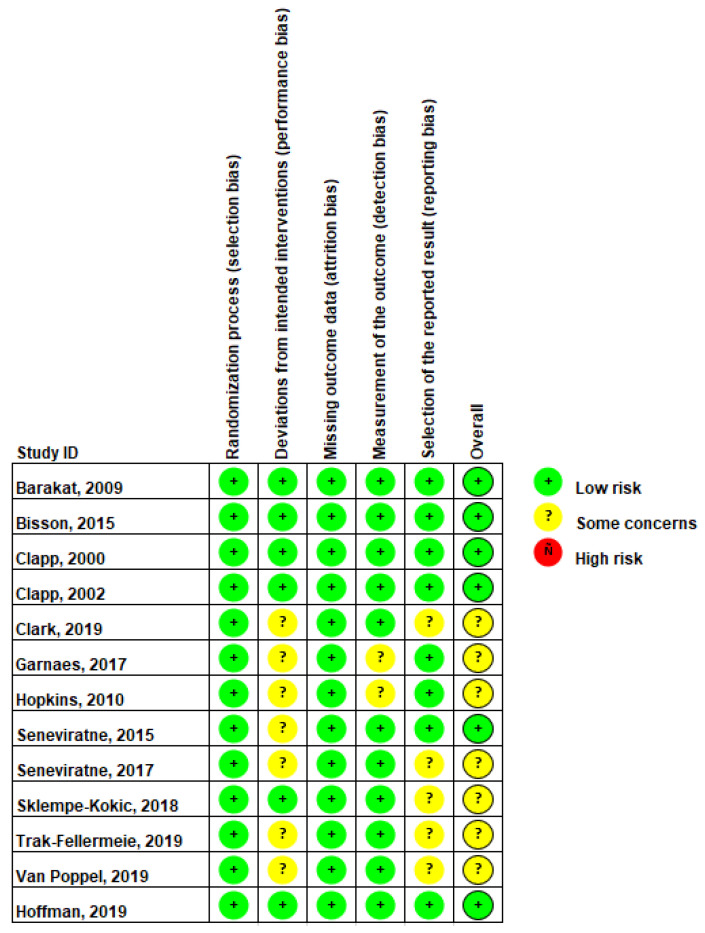
Randomized controlled trial risk of bias summary. Baraket, 2009 [17], Bisson, 2015 [18], Clapp, 2000 [19], Clapp, 2002 [20], Clark, 2019 [21], Garnaes, 2017 [22], Hopkins, 2010 [24], Seneviratne, 2015 [25], Seneviratne, 2017 [26], Sklempe-Kokic, 2018 [27], Trak-Fellermeie, 2019 [28], Van Poppel, 2019 [29], Hoffman, 2019 [23].

**Table 1 ijerph-19-07127-t001:** Characteristics of study populations.

Randomized Control Trial				
Authors, Year, Ref.	Country	Sample Size	Ethnicity	Pre-Pregnancy Weight Status
Barakat et al., 2009 [17]	Spain	160	Caucasian	Training 24.3 ± 0.5 kg/m^2^ Control 23.4 ± 0.5 kg/m^2^
Bisson et al., 2015 [18]	Canada	50	Caucasian (90%) Other	Training 34.6 ± 5.4 kg/m^2^ Control 33.9 ± 4.5 kg/m^2^
Clapp et al., 2000 [19]	USA	46	NA	Training 62.1 ± 1.1 kg Control 61.7 ± 1.3 kg
Clapp et al., 2002 [20]	USA	75	NA	Training * Lo–Hi 59.2 ± kg Mod–Mod 60.5 ± 1.1 kg Hi–Lo 58.9 ± 1.1 kg
Clark et al., 2019 [21]	USA	36	NA	Training 24.0 ± 5.2 kg/m^2^ Control 28.1 ± 8.0 kg/m^2^
Garnaes et al., 2017 [22]	Norway	74	Caucasian Other	Training 33.9 ± 3.8 kg/m^2^ Control 35.1 ± 4.6 kg/m^2^
Hoffman et al., 2019 [23]	Germany	2018	German (88%) Other	Training and Control 24.4 ± kg/m^2^
Hopkins et al., 2010 [24]	New Zealand	84	European (94%) Other	Training 25.5 ± 4.3 kg/m^2^ Control 25.4 ± 2.9 kg/m^2^
Seneviratne et al., 2017 [25]	New Zealand	72	Maori, Pacific Islander, NZ/European, Other	Training and Control 33.25 ± 5.4 kg/m^2^
Seneviratne et al., 2016 [26]	New Zealand	75	Maori, Pacific Islander, NZ/European, Other	Training and Control >25.0 kg/m^2^
Sklempe Kokic et al., 2018 [27]	Croatia	42	NA	Training 24.39 ± 4.9 kg/m^2^ Control 25.29 ± 4.7 kg/m^2^
Trak-Fellermeier et al., 2019 [28]	Puerto Rico	31	Black/AA (26%) Caucasian (22%) Other	Training 34.6 ± 8.0 kg/m^2^ Control 36.0 ± 7.0 kg/m^2^
Van Poppel, et al., 2019 [29]	Netherlands	334	Caucasian (65%) Other	Training 33.8 ± 3.9 kg/m^2^ Control 33.7 ± 3.7 kg/m^2^

* Lo–Hi consisted of moderate intensity weight-bearing exercise for 20 min, 5 days/week, through week 20 gestational age (GA), gradually increasing to 60 min, 5 days/week, by week 24 and maintaining that regimen until delivery; Mod–Mod consisted of moderate-intensity weight-bearing exercise for 40 min, 5 days/week, from week 8 until delivery; Hi–Lo consisted of moderate-intensity weight-bearing exercise for 60 min, 5 days/week, through week 20, gradually decreasing to 20 min, 5 days/week, by week 24 and maintaining that regimen until delivery.

**Table 2 ijerph-19-07127-t002:** Characteristics of study populations.

Cohort: Single-Arm Interventions				
Authors, Year, Ref.	Country	Sample Size	Ethnicity	Pre-Pregnancy Weight Status *
Badon et al., 2018 [30]	USA	3687	Caucasian (86%) Other	Normal (73%) Overweight/Obese (26%)
Badon et al., 2016 [31]	USA	3310	Caucasian (86%) Other	Normal (71%) Overweight/Obese (25%)
Bisson et al., 2017 [32]	Canada	104	Caucasian (96%) African American Other	23.7 ± 0.4 kg/m^2^
Collings et al., 2020 [33]	U.K.	6921	Caucasian (50%) Other	Active 24.7 kg/m^2^ Inactive 25.6 kg/m^2^
Dahly et al., 2018 [34]	Ireland	1754	Caucasian (98%) Other	Normal (65%) Overweight/Obese (35%)
Diaz et al., 2020 [35]	USA	209	Caucasian (85%) Other	26 kg/m^2^
Harrod et al., 2014 [36]	USA	826	Caucasian (53.4%) Black (16.7%) Hispanic Other	25.8 ± 0.4 kg/m^2^
Jones et al., 2020 [37]	USA	103	Caucasian (76%) Black Other	26.4 kg/m^2^
Joshi et al., 2005 [38]	India	770	Other	18.2 ± 0.3 kg/m^2^
Juhl et al., 2010 [39]	Denmark	79,692	Caucasian Other	Normal (68%) Overweight/Obese (27%)
Mudd et al., 2019 [40]	USA	37	Caucasian (81%) Other	Normal (62%) Overweight/Obese (38%)
Nagpal et al., 2018 [41]	Canada	61	Caucasian (88%) Other	Normal (69%) Overweight/Obese (31%)
Norris et al., 2017 [42]	Norway	1200	Caucasian (98%) Other	24 ± 0.5 kg/m^2^
Przybylowicz et al., 2014 [43]	Poland	510	NA	Normal (71%) Overweight/Obese (18.5%)
Rao et al., 2003 [44]	India	797	Indonesian Other	41.6 ± 5.1 kg
Watson et al., 2018 [45]	South Africa	130	Black	27.7 ± 5.2 kg/m^2^

* kilogram per meters squared (kg/m^2^).

**Table 3 ijerph-19-07127-t003:** Characteristics of study populations.

Case Control Studies				
Authors, Year, Ref.	Country	Sample Size	Ethnicity	Pre-Pregnancy Weight Status *
Tinius, Cahill, Strand, and Cade, 2016 [5]	USA	32	Caucasian (44%) African American (50%) Other (6%)	Active 34.0 ± 3.7 kg/m^2^ Inactive 36.3 ± 4.3 kg/m^2^
Clapp et al., 1998 [6]	USA	104	Caucasian	Exercise 60.0 ± 1.1 kg Control 60.4 ± 1.6 kg
Clapp and Capeless, 1990 [46]	USA	77	Caucasian	Exercise 57.7 ± 5.2 kg Control 58.1 ± 5.9 kg

* kilogram (kg); kilogram per meters squared (kg/m^2^).

**Table 4 ijerph-19-07127-t004:** Methods and timing of maternal physical activity data collection.

Randomized Control Trial					
Authors, Year, Ref.	PA Assessment Time Points *	Self-Reported	Objective	Total PA **	LTPA ***
Barakat et al., 2009 [17]	12–39 w	**-**	**X**	**X**	**-**
Bisson et al., 2015 [18]	14, 28, 36 w	**X**	**X**	**X**	**-**
Clapp et al., 2000 [19]	8–40 w	**-**	**X**	**X**	**-**
Clapp et al., 2002 [20]	8–40 w	**-**	**X**	**X**	**-**
Clark et al., 2019 [21]	16–36 w	**X**	**X**	**X**	**-**
Garnaes et al., 2017 [22]	12–40 w	**X**	**X**	**X**	**-**
Hoffman et al., 2019 [23]	<12, 29 w	**X**	**-**	**X**	**-**
Hopkins et al., 2010 [24]	20 w–delivery	**X**	**X**	**X**	**-**
Seneviratne et al., 2017 [25]	19–36 w	**X**	**X**	**X**	**-**
Seneviratne et al., 2016 [26]	20–36 w	**-**	**X**	**X**	**-**
Sklempe Kokic et al., 2018 [27]	30–40 w	**X**	**X**	**-**	**X**
Trak-Fellermeier et al., 2019 [28]	<16, 24–27, 35–36 w	**-**	**X**	**X**	**-**
Van Poppel, et al., 2019 [29]	<20–37 w	**X**	**X**	**-**	**X**

* physical activity (PA) assessment time points; weeks gestational age (w); weeks gestational age until delivery (w–delivery); ** total physical activity (PA); *** leisure time physical activity (LTPA).

**Table 5 ijerph-19-07127-t005:** Methods and timing of maternal physical activity data collection.

Cohort: Single-Arm Interventions					
Authors, Year, Ref.	PA Assessment Time Points *	Self-Reported	Objective	Total PA **	LTPA ***
Badon et al., 2018 [30]	15 w	**X**	**-**	**-**	**X**
Badon et al., 2016 [31]	15 w	**X**	**-**	**-**	**X**
Bisson et al., 2017 [32]	17, 36 w	**X**	**X**	**X**	**-**
Collings et al., 2020 [33]	26–28 w	**X**	**-**	**-**	**X**
Dahly et al., 2018 [34]	<15, 20 w	**X**	**-**	**-**	**X**
Diaz et al., 2020 [35]	<10 w	**-**	**X**	**X**	**-**
Harrod et al., 2014 [36]	17, 27 w, delivery	**X**	**-**	**X**	**X**
Jones et al., 2020 [37]	8–14, 20–23, 32–35 w	**-**	**X**	**X**	**X**
Joshi et al., 2005 [38]	18, 28 w	**X**	**-**	**X**	**-**
Juhl et al., 2010 [39]	16, 31 w	**X**	**-**	**-**	**X**
Mudd et al., 2019 [40]	Follow up at 4 y	**X**	**-**	**X**	**-**
Nagpal et al., 2018 [41]	24–28 w	**-**	**X**	**X**	**X**
Norris et al., 2017 [42]	<15, 20 w	**X**	**-**	**-**	**X**
Przybylowicz et al., 2014 [43]	Follow up at 1 month PP	**X**	**-**	**-**	**X**
Rao et al., 2003 [44]	18, 28 w	**X**	**-**	**-**	**X**
Watson et al., 2018 [45]	14–18, 29–33 w	**-**	**X**	**X**	**-**

* physical activity (PA) assessment time points; weeks gestational age (w); year (y); postpartum (PP); ** total physical activity (PA); *** leisure time physical activity (LTPA).

**Table 6 ijerph-19-07127-t006:** Methods and timing of maternal physical activity data collection.

Case Control Studies					
Authors, Year, Ref.	PA Assessment Time Points *	Self-Reported	Objective	Total PA **	LTPA ***
Tinius, Cahill, Strand, and Cade, 2016 [5]	32–37 w	**X**	**X**	**X**	**X**
Clapp et al., 1998 [6]	Follow up at 1 y	**-**	**X**	**X**	**-**
Clapp and Capeless, 1990 [46]	Preconception and throughout pregnancy	**-**	**X**	**X**	**-**

* physical activity (PA) assessment time points; weeks gestational age (w); year (y); ** total physical activity (PA); *** leisure time physical activity (LTPA).

**Table 7 ijerph-19-07127-t007:** Description of maternal physical activity, newborn body composition assessment and timing, and summary of key findings.

Randomized Control Trial					
Authors, Year, Ref.	Description of Maternal PA *	Newborn Body Comp. ASMT **	ASMT Timing ***	Key Findings	Body Comp.Diff. at Birth
Barakat et al., 2009 [17]	Resistance Training 3×/w	PI	At birth	No sig. assoc. between training women and infant outcomes. Control women’s pre-pregnancy weight positively assoc. with infant birthweight.	No
Bisson et al., 2015 [18]	Aerobics 3×/w; Resistance Training 3×/w; Accelerometer; PPAQ	SF	w/n 72 h of birth	No sig. diff. in infant outcomes between the training and control group.	No
Clapp et al., 2000 [19]	Aerobics 3×/w	SF PI TBEC	At birth TBEC at 5 days	Infants were sig. heavier, longer, and had more lean mass in the exercise group compared to the control group. All other infant outcomes not sig. diff.	No
Clapp et al., 2002 [20]	Aerobics 3–5×/w	SF PI TBEC	At birth TBEC at 5 days	Infants with moms who slowly increased exercise volume from first trimester (low) to third (high) were sig. smaller (smaller birth weight, smaller PI, body fat percent, fat mass and lean mass) than those infants whose moms started the first trimester with a high volume and decreased volume throughout pregnancy (high–low).	Yes
Clark et al., 2019 [21]	Aerobics 3×/w; MPAQ; HR monitoring	PI BMI Abd. Cir.	At birth	Pre-pregnancy PA levels sig. assoc. with PI and BMI. Infant head circumference in the exercise group significantly larger than infants in the control group.	No
Garnaes et al., 2017 [22]	Aerobics 3×/w; Resistance Training 2×/w	SF Abd. Cir.	At birth	No sig. diff. in infant outcomes between exercise and control group.	No
Hoffman et al., 2019 [23]	PPAQ	BMI	At birth	Women who were more active in late pregnancy sig. assoc. with a larger infant birthweight.	No
Hopkins et al., 2010 [24]	Aerobics 5×/w; HR monitoring	DXA PI BMI	w/n 48 h of birth DXA at 17 days	15 w exercise program during later pregnancy assoc. with reduced birth weights, but there were equal reductions in FM/FFM to account for the difference in weight, not just fat mass reductions, between the exercise and control groups. BMI sig. lower at birth in exercise vs. control; however, PI was not sig. diff.	No
Seneviratne et al., 2017 [25]	PPAQ; Aerobics 3–5×/w	DXA PI	w/n 2 w of birth	No sig. diff. in infant outcomes between the exercise and control group.	No
Seneviratne et al., 2016 [26]	Aerobics 3–5×/w	PI BMI	At birth	No sig. diff. in infant outcomes between exercise and control groups.	No
Sklempe Kokic et al., 2018 [27]	PPAQ; Aerobics 2×/w; Resistance Training 2×/w	PI	At birth	No sig. diff. in infant outcomes between the exercise and control group	No
Trak-Fellermeier et al., 2019 [28]	Accelerometer	SF PI Abd. Cir.	w/n one week	No sig. diff. in infant outcomes between the exercise and control group. SF data not presented	No
Van Poppel, et al., 2019 [29]	Aerobics 2×/w; PPAQ; Interview	SF Abd. Cir.	w/n 48 h of birth	No sig. diff. in infant outcomes between exercise and control group.	No

* Description of maternal physical activity (PA); times per week (x/w); pregnancy physical activity questionnaire (PPAQ); modifiable physical activity questionnaire (MPAQ); heart rate (HR); ** Newborn body composition (Comp.) assessment (ASMT); ponderal index (PI); skin fold (SF); total body electrical conductivity (TBEC); body mass index (BMI); dual energy X-ray absorptiometry (DXA) abdominal circumference (Abd. Cir.); *** Assessment (ASTM) timing; within (w/n); hours (h).

**Table 8 ijerph-19-07127-t008:** Description of maternal physical activity, newborn body composition assessment and timing, and summary of key findings.

Cohort: Single-Arm Interventions					
Authors, Year, Ref.	Description of Maternal PA *	Newborn Body Comp. ASMT **	ASMT Timing ***	Key Findings	Body Comp. Diff. at Birth
Badon et al., 2018 [30]	Focus Group Interview	PI	At birth	Yoga no assoc. with PI. Light to mod walking during pre-pregnancy and first trimester assoc. with infants’ greater PI. Longer durations/bouts of light to mod walking had larger increases of infant PI compared to women with no LTPA.	No
Badon et al., 2016 [31]	Interview	PI	At birth	Pre-pregnancy or first trimester LTPA not assoc. with PI.	No
Bisson et al., 2017 [32]	PPAQ Accelerometer	DXA	~12 days post birth	No sig. diff. in infant outcomes with any mod. PA across any trimester. Vig. PA in third trimester related to sig. lower infant BF and sig. smaller change in fat mass at 4 y.	No
Collings et al., 2020 [33]	GPPAQ	SF Abd. Cir.	w/n 24–72 h of birth	Higher mid-pregnancy PA levels for white British women were assoc. with smaller infant SF tricep and subscapular sum.	Yes
Dahly et al., 2018 [34]	Interview	PeaPod	w/n 48 h of birth	Mod PA during first trimester assoc. with lower infant BF.	Yes
Diaz et al., 2020 [35]	Accelerometer	PeaPod	w/n 2 w of birth	No assoc. found between PA measured by accelerometer and infant %BF. Maternal %BF was positively assoc. with both male and female infant %BF.	No
Harrod et al., 2014 [36]	PPAQ	PeaPod SF Abd. Cir.	w/n 72 h of birth	No sig. diff. for early and mid pregnancy total energy expenditure and infant fat mass/fat-free mass. Increasing late-pregnancy levels of TEE were sig. assoc. with decreased levels of infant adiposity; at extreme ends of total energy expenditure there was a sig. diff. in infant FM.	Yes
Jones et al., 2020 [37]	Accelerometer	PI Head Cir.	At birth	High vs. Low SED was assoc. with larger HC, longer BL, and lower PI; High MVPA was assoc. with smaller HC but was not assoc. with PI.	No
Joshi et al., 2005 [38]	PA Survey	SF Abd. Cir.	w/n 72 h of birth	No sig. diff. between PA levels and infant outcomes.	No
Juhl et al., 2010 [39]	Interview	PI Abd. Cir.	At birth	Sig. negative trend of exercise time during second trimester and infant abd. cir.	No
Mudd et al., 2019 [40]	Recall Questionnaire	PeaPod	w/n 2 w to 3 months of birth	Any mod PA in any trimester not assoc. with infant body comp. Vig. PA in third trimester sig. assoc. with lower infant BF and sig. smaller change in FM at 4 y.	No
Nagpal et al., 2018 [41]	Accelerometer	SF	w/n 24–48 h of birth	Sedentary time and MVPA not assoc. with infant BW or %BF.	No
Norris et al., 2017 [42]	Interview	PeaPod	w/n 72 h of birth	Pre-pregnancy and first trimester PA levels not assoc. with infant adiposity. Women with no PA between 15–20 w gestation were twice as likely to give birth to infants with adiposity above the 90th percentile.	No
Przybylowicz et al., 2014 [43]	PA Survey	PI	At birth	No sig. diff. in PI with any PA levels.	No
Rao et al., 2003 [44]	Focus Group Interview	SF Abd. Cir.	At birth	Maternal PA level not assoc. with abd. cir. or infant FM/FFM.	No
Watson et al., 2018 [45]	Accelerometer	PI	w/n 48 h of birth	No maternal PA assoc. with PI.	No

* Description of maternal physical activity (PA); pregnancy physical activity questionnaire (PPAQ); general practice physical activity questionnaire (GPPAQ); ** Newborn body composition (Comp.) assessment (ASMT); ponderal index (PI); skin fold (SF); head circumference (Cir.); dual energy X-ray absorptiometry (DXA); abdominal circumference (Abd. Cir.); *** Assessment (ASTM) timing; within (w/n); hours (h); weeks (w).

**Table 9 ijerph-19-07127-t009:** Description of maternal physical activity, newborn body composition assessment and timing, and summary of key findings.

Case Control Studies					
Authors, Year, Ref	Description of Maternal PA *	Newborn Body Comp. ASMT. **	ASMT Timing ***	Key Findings	Body Comp. Diff. @ Birth
Tinius, Cahill, Strand, & Cade, 2016 [5]	Health Survey Accelerometer	PeaPod SF	w/n 48 h of birth	No sig. diff. in infant outcomes assoc. with maternal PA levels	No
Clapp et al., 1998 [6]	Aerobics 3x/w HR monitoring	SF Abd Cir. TBEC	w/n 24 h of birth	Exercise group assoc. with sig. lower infant %BF at birth but not sig. at one year	No
Clapp & Capeless, 1990 [46]	Aerobics 3x/w HR monitoring	SF PI	<2 h of birth	Exercise group had infants with sig. smaller FM & %BF. Duration of exercise but not type of exercise assoc. with sig. smaller PI	Yes

* Description of maternal physical activity (PA); heart rate (HR); times per week (x/w); ** Newborn body composition (Comp.) assessment (ASMT); ponderal index (PI); skin fold (SF); total body electrical conductivity (TBEC); abdominal circumference (Abd. Cir.); *** Assessment (ASTM) timing; within (w/n); hours (h).

## Data Availability

Not applicable.

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
