# Peer review of "Physical Activity during Pregnancy and Newborn Body Composition: A Systematic Review"

_ijerph, 2022, doi:10.3390/ijerph19127127_

Round 1
Reviewer 1 Report
Dear Authors,
Let me congratulate you on picking up so complicated topic! Although there are many points that can be discusssed, starting from the neonatal body compositions as a marker of neonate's well-being which might be somehow controversial, you performed the study and analysis perfectly. Of course, one might wish for longer or more in-depth discussion, although the paper has its limitations and in this form it cannot be required.
The only topic I suggest you discussed is a pre-pregnancy and pregnancy BMI, as it might influence neonatal body composition- I would suggest to discuss it at least a little bit, maybe by adding a short commentary at the end of the discussion section
Besides, I have only some minor comments:
line 35- the word "overwhelmingly" is too controversial, I would advise to change it or just skip it
l. 51-52: "(e.g., a 3500-gram newborn could be perfectly 51
grown, too small, or too big depending on its genetic potential)"- it is obvious, but the sentence is so obvious and simple that it just does not deserve to be in your eminent paper- erase it, please:)
l. 61 over and 10+ do not need to come together- stay with over only
Best regards
Author Response
Point 1: Dear Authors,
Let me congratulate you on picking up so complicated topic! Although there are many points that can be discussed, starting from the neonatal body compositions as a marker of neonate's well-being which might be somehow controversial, you performed the study and analysis perfectly. Of course, one might wish for longer or more in-depth discussion, although the paper has its limitations and in this form it cannot be required.
Response 1: Thank you very much for the kind words.
Point 2: The only topic I suggest you discussed is a pre-pregnancy and pregnancy BMI, as it might influence neonatal body composition- I would suggest to discuss it at least a little bit, maybe by adding a short commentary at the end of the discussion section
Response 2: Thank you for this comment. We have added additional information addressing the fact that others have found an association between maternal weight status and neonatal adiposity and expanded on the findings of Barakat et. Al. The last paragraph of the Discussion section now reads:
Specifically, Barakat et.al. found that maternal body weight was positively and significantly associate with birthweight in the control group, but there was no association in the exercise training group. Others have reported that maternal pre-pregnancy BMI and gestational weight gain are positively and independently associated with neonatal adiposity [59]. It is possible that the impact of maternal physical activity on neonatal adiposity may differ according to maternal pre-pregnancy weight status and/or gestational weight gain status.
Point 3: Besides, I have only some minor comments:
line 35- the word "overwhelmingly" is too controversial, I would advise to change it or just skip it
Response 3: This word has been removed. The sentence now reads: Scientific evidence has demonstrated the safety and efficacy of exercise during the perinatal period[1-5].
Point 4: l. 51-52: "(e.g., a 3500-gram newborn could be perfectly 51
grown, too small, or too big depending on its genetic potential)"- it is obvious, but the sentence is so obvious and simple that it just does not deserve to be in your eminent paper- erase it, please:)
Response 4: Thank you for this comment. We have removed the verbiage in parentheses. The sentence now reads: It also does not account for how much an infant “should have weighed” based on their length and genetic potential [11].
Point 5: l. 61 over and 10+ do not need to come together- stay with over only
Best regards
Response 5: We have removed that addition sign so that the beginning of the sentence now reads: Over the past 10 years,
Reviewer 2 Report
Dear authors,
First of all, I 'd like to congratulate you on the very interesting, innovative and with a methodology applied accurately.
I consider your manuscript of real interest to the readers , underlined that these review' results add to the consensus that health care providers should continue to encourage PA among patients while dispelling.
The only thing that I suggest you is to add to each table the foodnotes below.
Author Response
Response to Reviewer 2 Comments
Point 1:
Dear authors,
First of all, I 'd like to congratulate you on the very interesting, innovative and with a methodology applied accurately.
I consider your manuscript of real interest to the readers , underlined that these review' results add to the consensus that health care providers should continue to encourage PA among patients while dispelling.
The only thing that I suggest you is to add to each table the foodnotes below.
Response 1: Thank you very much for the kind words. We have added footnotes to each table in the manuscript.
Reviewer 3 Report
Physical Activity during Pregnancy and Newborn Body Composition: A Systematic Review
The manuscript is a systematic review presenting the association of maternal physical activity during pregnancy on neonatal body composition assessed between birth and two weeks of age.
The manuscript presents the results of the literature. The methodology is well presented. The studies are grouped into randomized control trials, cohort: single-arm interventions, and case-control studies.
The results are presented clearly in the tables. However, I have one remark.
Please provide an explanation of abbreviations under each table. This would help the manuscript to be better understood by the reader.
The discussion is well-grounded in the literature.
No further comments. Congratulations to the Authors!
Author Response
Response to Reviewer 3 Comments
Point 1:
The manuscript is a systematic review presenting the association of maternal physical activity during pregnancy on neonatal body composition assessed between birth and two weeks of age.
The manuscript presents the results of the literature. The methodology is well presented. The studies are grouped into randomized control trials, cohort: single-arm interventions, and case-control studies.
The results are presented clearly in the tables. However, I have one remark.
Please provide an explanation of abbreviations under each table. This would help the manuscript to be better understood by the reader.
The discussion is well-grounded in the literature.
No further comments. Congratulations to the Authors!
Response 1: Thank you very much for the kind words. We have added footnotes, explaining the abbreviations within each table, to each table in the manuscript.